# Forearm Fracture Nonunion with and without Bone Loss: An Overview of Adult and Child Populations

**DOI:** 10.3390/jcm11144106

**Published:** 2022-07-15

**Authors:** Sara Dimartino, Vito Pavone, Michela Carnazza, Enrica Rosalia Cuffaro, Francesco Sergi, Gianluca Testa

**Affiliations:** Department of General Surgery and Medical Surgical Specialties, Section of Orthopaedics and Traumatology, University Hospital Policlinico “Rodolico-San Marco”, University of Catania, 95123 Catania, Italy; saradimartino1@gmail.com (S.D.); vitopavone@hotmail.com (V.P.); michela.carnazza@libero.it (M.C.); enricacuffaro@outlook.it (E.R.C.); fsrgbl33@gmail.com (F.S.)

**Keywords:** forearm, nonunion, epidemiology, risk factor, children, treatment, external fixation, bone graft

## Abstract

Nonunion occurs in 2–10% of all forearm fractures due to different mechanical and biological factors, patient characteristics, and surgeon-dependent causes. It is a condition that causes functional and psychosocial disability for the patient because it is a unique anatomical segment in which all the bones and structures involved embody a complex functional unit; therefore, it is a challenge for the orthopedic surgeon. The ultimate goal of the care of these patients is the restoration of function and limitations related to impairment and disability. The aim of this review is to provide an extended description of nonunion forearm fractures, related risk factors, diagnosis, classification systems, and the available evidence for different types of treatment as a tool to better manage this pathology.

## 1. Introduction

Forearm nonunion represents a challenge for orthopedic surgeons both in terms of diagnosis and treatment, sometimes requiring reconstruction skills and good patient compliance because of the difficult treatment and long follow-up. The Food and Drug Administration (FDA) describes nonunion as a fractured bone that has not healed within nine months after trauma and shows no signs of progression of healing on radiographs over the course of three consecutive months [1]. In addition to this definition, nonunion has no chance of healing without additional operation. In the pediatric population, it is defined as an absence of fracture healing progression as shown on sequential radiographs or no evidence of healing by more than 10 weeks following the injury. Nonunion occurs in 2–10% of all forearm fractures, with or without infection and bone loss. A peak incidence has been observed in the age range between the ages of 35 and 44 and decreases thereafter [2], with a higher incidence in women after menopause. The most common fracture in the pediatric population is a forearm fracture [3], but nonunion is an uncommon complication after surgical treatment of displaced bones and has been described in only a few cases in the literature. Forearm nonunion is a condition that causes functional and psychosocial disability for the patient, resulting in the lowest health-related quality of life compared with other long bone nonunion and disease such as type 1 diabetes mellitus, stroke, and acquired immunodeficiency syndrome [4]. Forearm nonunion consists of different mechanical and biological factors: the type of fracture, patient characteristics, and surgeon-dependent causes. The evaluation of a suspected forearm nonunion includes medical history, laboratory tests, and clinical and instrumental factors. Successful treatment is the restoration of function and limitations related to impairment and disability. The aim of this review is to provide a correct forearm nonunion characterization with related risk factors, diagnosis, classification systems, and management with available evidence for different types of treatment.

## 2. Evaluation

### 2.1. Risk Factors

Forearm nonunion is due to failure of bone healing and is caused by many factors. In clinical practice, these factors are divided into mechanical and biological ones [5,6], as shown in Table 1, and in the literature, they are divided into general and local risk factors [7,8], as shown in Table 2.

### 2.2. History and Physical Examination

The assessment of a suspected forearm nonunion should start with the remote pathological history of the patient, together with investigation of the preliminary elements to guide the diagnostic path: the presence of risk factors, mechanism of trauma, details of prior surgical procedures, distress with weight bearing, and factors of infection. The physical examination should detect differences with the contralateral side in terms of the presence of shortening, range of movement, and lower grip strength; and assess the skin covering the nonunion and its mobility. Tenderness on palpation and preternatural mobility at the fracture site may result. The latter could represent a clinical sign of incomplete bone healing that could be associated with pain, poor functionality, and mechanical instability. The clinical examination should also assess the presence of any deformity, soft tissue, limb vascularity, and arm muscle circumference as an indicator of nutritional status. It is important to diagnose aseptic versus septic nonunion [9] because if sepsis is present, the management varies significantly.

### 2.3. Imaging

The diagnostic evaluation continues with radiographic aspects through the anteroposterior, lateral, and oblique views of the injured bone and adjacent joints in addition to the same view of the original fracture. Radiological signs include variable bone callus presentation, fracture stumps sclerosis with persistence of a fracture line, and the presence of a defect or a deformity. Radiography of the contralateral side may be useful as it may outline shortening and concurrent malunions or normal characteristics for the patient. In addition, computed tomography (CT) scans are frequently used in current practice (Figure 1) to identify unhealed fractures, which is useful for anticipating negative evolution of the reparative process.

Contrast-enhanced ultrasound (CEUS), also with the adjunct of Doppler, is helpful for investigating the presence of vessels from the perspective of vascularized flaps. Enhanced magnetic resonance with dynamic contrast (DCE-MRI) has been proposed to analyze the infection status and perfusion of a nonunion [10]. 

### 2.4. Laboratory Analysis

Additional diagnostic insights include an evaluation of inflammation markers (white blood cells, erythrocyte sedimentation rate, and C-reactive protein (CRP)); biochemical elements, such as liver function, thyroid and parathyroid, calcium, and vitamin D; in addition to multiple samples for culture examination before any antibiotic prophylaxis. Brinker et al. [11] found that the most common abnormality was vitamin D deficiency. Criteria indicative of an infection are a white blood cell count greater than 11,000 × 10^9^, an erythrocyte sedimentation rate >30 mm/h, and a CRP level >1.0 mg/dL; three positive tests have a predictive infection value of 100% [12] These tests are useful for an initial distinction between septic and aseptic nonunion.

### 2.5. Characterization

The most utilized nonunion classification system was proposed by Weber and Cech, describing atrophic, oligotrophic, and hypertrophic based on callus formation as shown on the radiograph. Hypertrophic nonunion, usually resulting from insufficient fracture stabilization and showing adequate vascularization, is marked by extensive callus formation with a horse-shoe or elephant-foot radiographic configuration (Figure 2).

Atrophic nonunion, resulting from dysfunction in biological activity with insufficient vascularization but adequate fracture stabilization, typically radiologically shows minimal callus around a fibrous tissue-filled fracture gap. Oligotrophic nonunion shows some of the radiographic and biologic features of each type and typically presents biologic potential for healing with no initiation of healing and minimal to no callus formation. The AO (Arbeitsgemeinschaft für Osteosynthesefragen) classification scheme adds the concept of pseudoarthrosis in cases in which the formation of a false joint due to the persistence of movement at the site of fracture occurs [8]. Unfortunately, there is no specific classification for forearm nonunion and regarding future directions, it would be useful to create a dedicated one that is based on the following: the score obtained from the Weber and Cech, the sites of nonunion (distal, diaphysis, or proximal), whether it affects one or both bones, pediatric or adult, state of infection, with vascular deficiency, and with or without bone loss.

## 3. Pediatric Treatment

Forearm fractures are usual in children: 34% of cases occur in children at ages ranging from 5 to 14 years [13]. Nonunion is prevalent in children older than 10 years as compared to children under 10 years due to lesser bone remodeling potential in older children. In particular, it depends on open reduction and wide bone exposure, poor fixation, an inadequate period of immobilization (<8 weeks), and early hardware removal (<3 months). Generally, in children, the ulna is more involved than the radius [14]. Trauma in the middle third of the ulna, also called the “water-shed zone”, is critical because the intraosseous circulation may be compromised, which may invalidate bone healing [15,16]. The distal third of the radius is involved due to damage of the pronator quadratus vascularization. In children’s forearm fracture nonunion, it is important to evaluate the type of treatment that has been carried out previously. Ogonda et al. [17] analyzed elastic stable intramedullary nailing (ESIN) fixation in both-bone forearm fractures and showed that the frequency of ulna nonunion was higher in anterograde nailing than in retrograde, and radius nonunion was less frequent due to compression of the fracture line. However, Yaradilmis et al. [18] demonstrated that intramedullary nailing is minimally invasive and provides biological fixation. In addition, plate-screw nonunion fixation depends on wide soft tissue dissection and stripping of the bone periosteum to provide adequate exposure [19]. Metabolic disorders should be corrected to stimulate bone healing; if a lack of vitamin D is present, supplementation should be provided to promote better consolidation. Loose et al. [20] showed that a conservative approach could be adopted in asymptomatic patients; however, if a young patient is considered symptomatic when they present with an angular deformity, functional deficit or movement restriction, and pain, it is important to assess surgery. The best operative management consists of osteosynthesis with or without bone grafting (Figure 3).

First, fibrous tissue is removed followed by decortication and opening of the medullary canal. In a septic nonunion, it is important to analyze bioptic samples. Second, bone grafting with an allograft or autograft has been performed. Stabilization of a nonunion is achieved with tubular plates, dynamic compression plates or locking compression plates (DCPs or LCPs), Kirschner wires, rush rods, or an external fixator. After surgery, cast immobilization is required. An algorithm from the authors is illustrated in Figure 4.

Surgical treatment of pediatric forearm nonunion provides satisfactory outcomes but is associated with sequelae and residual functional disability with several complications, such as radio-ulnar fusion, radial nerve palsy, myositis ossificans at the ulna, and olecranon bursitis with elbow stiffness.

## 4. Adult Forearm Nonunion Surgical Treatment 

The goal of surgical management of forearm nonunion is to recover the proper bone length, anatomy, and functionality, and remove pain. Achieving a successful outcome in the management of forearm nonunion treatment requires optimization of both the patient’s biological state and the stability of the nonunion site. Conservative treatment is only for special patients who are not suitable candidates for surgical treatments. The gold standard for managing septic forearm nonunion is a staged approach [21] to control infection via the use of a debridement, antibiotic spacer, and cultures followed by definitive surgery while for the aseptic situation, a single-stage treatment is used [22]. Management of nonunion forearm fracture in adults includes different types of surgical treatment such as bone grafting with compression with plates, the Masquelet technique, and external fixation. Compression plating and autologous bone grafting have mainly been regarded as the keystone of nonunion treatment. Both mechanical and biologic failure are corrected by restoring stability (by compression plating) and introducing osteoinductive and/or osteoconductive agents (by autologous bone graft). In oligotrophic or atrophic nonunion, bone grafting is necessary to fill the gap between the two ends. Ring et al. [23] obtained a 100% union rate with an autogenous iliac crest bone graft and 3.5 mm plate and screw for a forearm nonunion.

### 4.1. Bone Grafting

Different kinds of bone grafts exist, but autologous cancellous bone is still being considered as the gold standard to provide a biological stimulus for the consolidation process. A bone graft can be cancellous, corticocancellous, and/or vascularized. Cancellous grafts can derive from the iliac crest, distal radius, olecranon, lateral epicondyle, tibial metaphysis, or reamer/irrigator/aspirator (RIA) of the femur. A corticocancellous graft includes the tricortical iliac crest, free fibula, and medial femoral condyle. The size of the gap may be influenced by the choice of being vascularized or not. A vascularized bone graft is used for gaps greater than 5 to 6 cm; in particular, the free fibula is a great option as shown for forearm nonunion by Adani et al. [24]. An autologous graft has the advantage of being osteogenic, osteoinductive, and osteoconductive; it is biologically superior to homologous grafts. However, the disadvantages include donor site morbidity, pain, and limited supplies. For forearm nonunion, the best utilized autograft options include the iliac crest bone graft (ICBG) as shown by Regan et al. [25] and RIA of the femur. With the RIA technique, a graft of greater volume than ICBG is obtained as shown by Dawson et al. [26] and RIA is superior in improving the cost-benefit when surgical time is considered and lowering donor-site pain. On the other hand, significant blood loss may occur. Compared with an autograft, an allograft prevents donor-site morbidity and may reduce the surgical time. Allografts can contribute to bone reconstruction thanks to their osteoinductive and osteoconductive properties but require a vital environment to be effective. Moreover, the use of structural allografts can be complicated by infection, incomplete remodeling, fracture, and disease transmission. Vascularized bone grafts can be obtained from the fibula, iliac crest, rib, radius, ulna, scapula, femur, humerus, and pubic bone or metatarsals, among other sites. At present, the most frequently applied technique for bone defects > 5–6 cm in the septic non-union of one or both forearm bones is the free vascularized fibular graft (FVFG). Because of its anatomical and mechanical characteristics, it is an excellent graft for the reconstruction of forearm bone defects. The fibula has a diameter similar to that of the forearm bones, the morbidity of the donor zone is minimal, and the length available for extraction is usually sufficient [27]. One of the main advantages of FVFG is that in a single surgical intervention, it enables reconstruction of one or both forearm bones in addition to coverage of any soft tissue defects in patients with complex trauma or in infected areas with poor vascularization [28]. Vascularized bone grafts have an improved rate of survival in a poorly vascularized bed. Gan et al. [29] demonstrated for bone defects in pseudarthroses of the forearm that a vascularized fibular graft is an optimal option with fracture healing. Among its advantages, vascularized bone grafting facilitates the provision of nutrients to the deep structures of the graft and enables stable osteosynthesis, thus allowing prompt mobilization of the limb and promoting functional recovery [30]. A potential disadvantage is that the operation requires microvascular surgical skills.

### 4.2. Induced Membrane Technique

The induced membrane technique, also known as the Masquelet technique [31], is applicable under both aseptic and septic conditions leading to substantial bone loss and requires no advanced skills in microvascular surgery. The technique involves a two-stage procedure to restore the bone defect. Ma et al. [32] studied the induced membrane technique for infected forearm nonunion in 32 patients who haled without recurrent infection or loosening of internal fixation, finding it to be an effective solution. Walker et al. [33] demonstrated successful use of this method in forearm nonunion for defects up to 5.4 cm in size. Pachera et al. [34] reported a case of a 53-year-old patient with a left forearm deformity due to an atrophic nonunion of the ulna and a malunion of the radius, which was successfully managed with the use of the Masquelet technique associated with a corrective osteotomy of the radius, performed with the aid of a 3D model. Potential problems with the Masquelet technique include loosening of the fixation implant, infection, fracture through the graft, and bone resorption.

### 4.3. Ilizarov with Bone Transport

Ilizarov fixation with bone transport is a viable treatment option for atrophic forearm nonunion and is particularly indicated in cases of significant soft tissue damage or nonunion with infection. The Ilizarov methodology allows bridging of bone losses (caused by osteogenesis) with bone transport and provides stable fixation without implantation of permanent foreign bodies, thus permitting wrist and elbow movement. This technique, therefore, allows for immediate therapy of the hand, wrist, and elbow in addition to early use of the extremities during the activities of daily living. Moreover, circular systems can be used to correct complex multiplanar deformities in small areas of soft tissue defects and immediate mobilization. Its disadvantages are the long duration of external osteosynthesis materials, the frequency of pin-tract infections, and the pain accompanying the transport. Zhu et al. [35] studied the effectiveness of the Ilizarov technology for the treatment of infected forearm nonunion with satisfactory clinical results, finding radical debridement is the key to controlling bone infection. Orzechowski et al. [36] demonstrated that the Ilizarov is the method of choice in the treatment of forearm nonunion with concomitant shortening and axis deformity. Liu et al. [37] treated 12 patients with diaphyseal forearm bone defects caused by infection with bone transport using a monolateral external fixator and all patients achieved infection-free union. 

## 5. Conclusions

Forearm nonunion is an uncommon but complex condition problem, with countless different presentations. In a future prospective, it would be useful to create a specific classification system to guide the right management. The question: “Why did the fracture not heal?” should be addressed by the surgeon to investigate risk factors, correct the metabolic abnormalities, and study the nonunion imaging characteristics to optimize the patient’s biology and stability at the affected site. What kind of graft should be used? This depends on the presence of infection, patient characteristics, and the size of the defect: a cancellous graft should be used when there is cortical contact while vascularized free fibula should be preferred for defects larger than 5–6 cm. Different surgery treatments have been used successfully and future studies should investigate the role of 3D printing in the pre-operatory planning, its intraoperatory advantages, and its role in bone grafting selection. This paper presents some limitations related to its narrative nature. In fact, a quality evaluation of the literature was carried out; therefore, a statistical comparison between the references was not possible. For this reason, it appears to be difficult to confront different studies having also a low grade of evidence. Therefore, we strongly support the need to design new studies in this direction. 

## Figures and Tables

**Figure 1 jcm-11-04106-f001:**
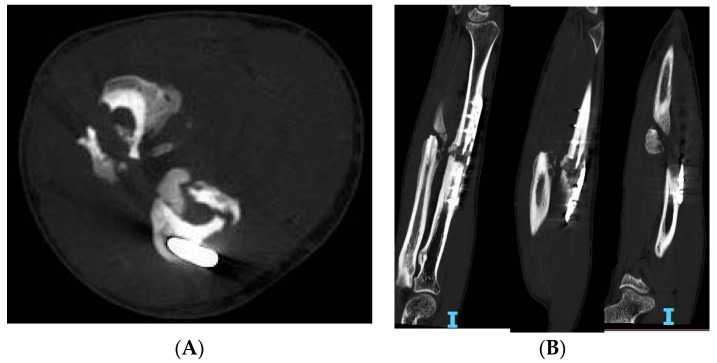
Forearm nonunion CT scans of a 43-year-old patient. (**A**) Coronal view, (**B**) Sagittal view.

**Figure 2 jcm-11-04106-f002:**
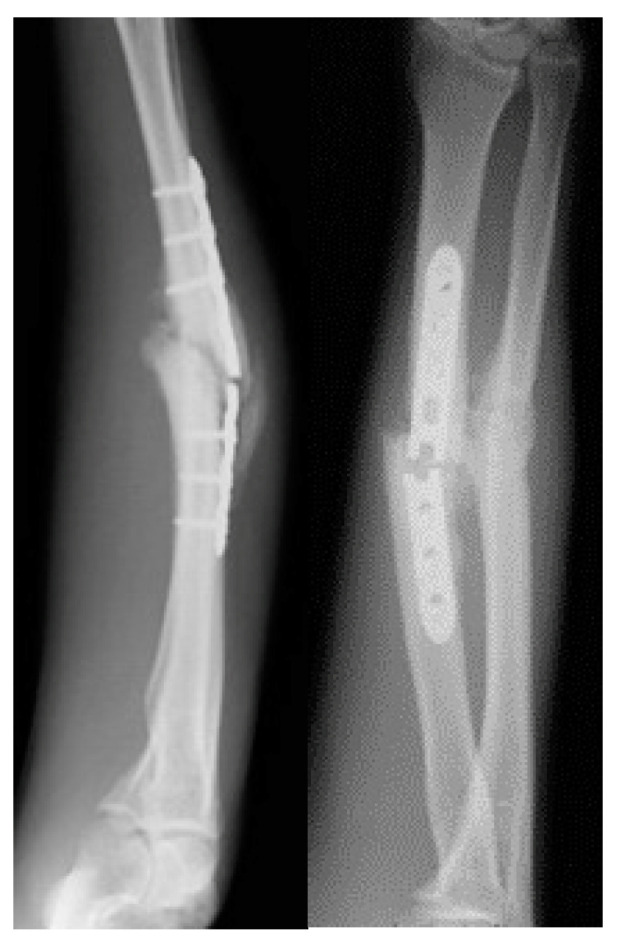
A 38-year-old patient presented a hypertrophic symptomatic nonunion with a broken plate after 9 months.

**Figure 3 jcm-11-04106-f003:**
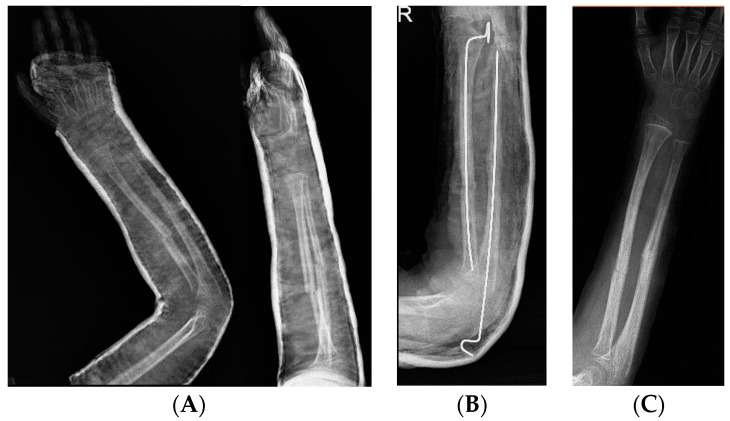
(**A**): Five-year-old patient treated with nonoperative treatment; after 4 months, they presented a displaced oligotrophic nonunion with pain and range of motion (ROM) deficit. (**B**): He was treated by cruentation of the fracture’s sites, reduction, stabilization with k wires, and cast. (**C**): Post k-wires removal.

**Figure 4 jcm-11-04106-f004:**
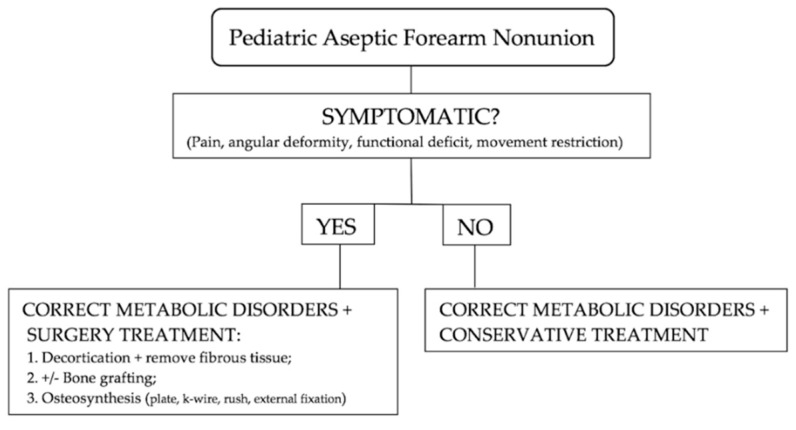
Treatment algorithm for pediatric aseptic forearm nonunion.

**Table 1 jcm-11-04106-t001:** Mechanical and biological factors.

**Mechanical Factors** [5,6]
Insufficient immobilization Nonoperative treatment Poor internal or external fixation
Excessive motion at the fracture site Malreduction or an unbalanced osteosynthesis system
**Biological Factors** [5,6]
Local: bone defects, open fracture, infection, soft tissue injuries, segmental fractures, pathological and comminuted fractures, and inter-fragmentary gap Systemic: neuropathy, diabetes, chronic smoking, chronic alcoholism, drugs, and radiation therapy

**Table 2 jcm-11-04106-t002:** General and local risk factors.

**GENERAL RISK FACTORS**
Gender [8] Age [6] Poor protein diet [6] Calcium and phosphorus deficit [8] Lack of vitamin D [7] Osteoporosis [8] Diabetes [7] Low muscular mass Alcohol [8] Smoking [6] Drugs (NSAIDS, opioid) [6] Infection [7,8] Radiation therapy [6] Neuropathies Genetic disorders (osteogenesis imperfecta)
**LOCAL RISK FACTORS**
Fracture type [7,8] Mechanism of injury [7,8] Exposure [6,7] Biological damage during the first surgery [6,8] Surgical techniques during fracture synthesis [7,8]

## Data Availability

Not applicable.

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
