# Peer review of "Forearm Fracture Nonunion with and without Bone Loss: An Overview of Adult and Child Populations"

_jcm, 2022, doi:10.3390/jcm11144106_

Round 1

Reviewer 1 Report

We are facing a revision, but a very simple revision. I believe that the possibilities of this work could have been much higher. 

When the authors propose an algorithm, it should have more bibliographic support. When is a pediatric fracture considered symptomatic?

In the adult, what would STAGE 1 be? Is metabolic study and control not necessary if there are signs of infection?

Finally, don't the authors believe that the study has some limitations? Should they not be included in the conclusions?

Author Response

Letter to Reviewer #1

First of all, we would all like to say thank you for your helpful correction and support during this entire time so far.

About the paper, we have reviewed the highly similar sentences according to the iThenticate report in the attachment. We eliminated the big general explanations section since we believe that who may read the paper, would already knows such terms (like what a Masquelet is).

Reply to Reviewer #1

  1. Q: When the authors propose an algorithm, it should have more bibliographic support. When is a pediatric fracture considered symptomatic?

A: We have also cited the Loose’s at al study [20] which to our knowledge is the largest reported series of such patients in the literature. We explained that a pediatric fracture could be considered symptomatic when it produces pain, presenting angular deformity, functional deficit or movement restriction.

  1. Q: In the adult, what would STAGE 1 be? Is metabolic study and control not necessary if there are signs of infection?

A: We pointed out clearlier the different stage approaches with STAGE 1 and STAGE 2, we also mentioned the correction of metabolic disorders in the infection nonunion management.

  1. Q: Finally, don't the authors believe that the study has some limitations? Should they not be included in the conclusions?

A: Of course the study has some limitations. We apologize for not mentioning them previously, so we added them to the conclusion.

We hope you appreciate our job and that you may now enjoy our work even more. At the same time, we remain at your disposal for any further modification and adjustments that you may like to suggest us.

Best Regards,  

The team.

Reviewer 2 Report

The authors need to address these comments.

1.  All abbreviations need to be given either at the beginning or at the end of the manuscript, like, According to the Food and Drug Administration (FDA), computed tomography (CT), etc.,

2.  Suitable references need to be given in table 1 and table 2.

3.  Figure 1. Forearm nonunion CT scans of a 43-year-old patient. All the views [Sagittal, Coronal] have to be mentioned, and also the figure should be the same size to maintain the uniformity

4.  Materials used for the metal plates have to be added and also the significance of this with nonunion can be correlated.

5.  Figure 5., Figure 6. The fonts are not matching with the manuscript text. Also, the quality of the figure has to be increased.

6.  Limitations of the current review have to be stated.

7.  Recent works have to be stated in the manuscript.

8.  The relations between each chapter have to be established.

Author Response

Letter to Reviewer #2

First of all, we would all like to say thank you for your helpful correction and support during this entire time so far.

About the paper, we have reviewed the highly similar sentences according to the iThenticate report in the attachment. We eliminated the big general explanations section since we believe that who may read the paper, would already knows such terms (like what a Masquelet is).

Reply to Reviewer #2

  1. Q: All abbreviations need to be given either at the beginning or at the end of the manuscript, like, According to the Food and Drug Administration (FDA), computed tomography (CT), etc.,

A: Thank you, that was corrected.

  1. Q: Suitable references need to be given in table 1 and table 2.

A: Thank you, we added the references for risk factors.

  1. Q: Figure 1. Forearm nonunion CT scans of a 43-year-old patient. All the views [Sagittal, Coronal] have to be mentioned, and also the figure should be the same size to maintain the uniformity

A: Thank you, that was corrected.

  1. Q: Materials used for the metal plates have to be added and also the significance of this with nonunion can be correlated.

A: Unfortunally the role of each material that compose the metal plates isn’t mentioned in literature, but the plate has an important role in terms of biological damage during the first surgery and unbalanced or instable osteosynthesis.

  1. Q: Figure 5., Figure 6. The fonts are not matching with the manuscript text. Also, the quality of the figure has to be increased.

A: Thank you. Yes, we did some modifications to the fonts in order to match better the manuscript text.

  1. Q: Limitations of the current review have to be stated.

A: Of course the study has some limitations. We apologize for not mentioning them previously, so we added them to the conclusion.

  1. Q: Recent works have to be stated in the manuscript.

A: We apologize however we reported the latest works that we were able to find in the literature.

  1. Q: The relations between each chapter have to be established.

A: Paper begins with an introduction, followed by an evaluation of the clinical and strumental aspects. On this basis we also discuss about the different treatment options and finally the conclusion sums up our final thoughts.

We hope you appreciate our job and that you may now enjoy our work even more. At the same time, we remain at your disposal for any further modification and adjustments that you may like to suggest us.

Best Regards,  

The team.

Round 2

Reviewer 1 Report

After the corrections made by the authors, I believe that the work is correct.

Reviewer 2 Report

Good Work

This manuscript is a resubmission of an earlier submission. The following is a list of the peer review reports and author responses from that submission.

Round 1

Reviewer 1 Report

Review of manuscript: „Nonunion forearm fracture: an overview of adult and child populations” J. Clin. Med. 2021, 10, x FOR PEER REVIEW.

The manuscript is very interesting and practical.  The authors show that nonunion is one of the most frequent complications of fracture that causes failure of bones to heal; nonunion occurs in 2%–10% of all forearm fractures. Surgical and additional treatments for children and adults are described. Forearm nonunion is a condition that causes functional and psychosocial disability for the patient. The authors also emphasized that the ultimate goal of the care of these patients is the restoration of function and limitations related to impairment and disability.

The work is very well structured, divided into clear paragraphs, includes definitions of nonunion according to the Food and Drug Administration.

The author showed that new biological stimulation techniques with the aim of delivering osteogenic components have been developed.

I have only a minor comments:

  1. It would also be useful to include a graphical abstract at the very beginning.
  2. In the pathophysiology section, despite 2 tables, you can additionally make a figure which will even more clearly show the factors leading to the development of nonunion.
  3. It would be interesting to include radiological images (CT, CEUS, DCE-MRI) showing forearm nonunion in children and adults.
  4. In the treatment paragraph, I suggest including 2 tables - 1 with surgical and 2 with non-operative treatment methods with attached citations.

According to my opinion, this manuscript is suitable for publication after this revision, which will increase the value of the presented work.

Author Response

Thank you for reviewing this article. We are very proud for your appreciation.

Following your suggestion, we have added a figure (Figure 1) that clearly shows the factors leading to the development of nonunion. We have included radiological images (ct ed rx) and two tables, one with surgical treatment and the second with non-operative treatment. 

Reviewer 2 Report

This paper does not present ny novelty in the field of orthopedics. The paper is structured as a course for medical students or young rezidents. The paper is not in the field proposed by Special Issue Orthopedic Treatment of Diseases and Fractures in Elderly.

Author Response

Dear Reviewer, thank you for reading our review on forearm nonunion. 

We are sorry that you have not appreciate our summary of literature on this topic, that represents a common complication of these fractures.

We kindly require you to suggest any possible improvements or criticisms, that determined your opinion on this manuscript. 

Best regards,

Gianluca Testa